# New Evaluation of Isoflavone Exposure in the French Population

**DOI:** 10.3390/nu11102308

**Published:** 2019-09-28

**Authors:** Alexandre Lee, Laetitia Beaubernard, Valérie Lamothe, Catherine Bennetau-Pelissero

**Affiliations:** 1Bordeaux Sciences Agro, F-33175 Gradignan, France; alexandre.lee@agro-bordeaux.fr (A.L.); l.beaubernard@gmail.com (L.B.); valerie.lamothe@agro-bordeaux.fr (V.L.); 2Pharmacy Faculty, University of Bordeaux, F-33077 Bordeaux, France

**Keywords:** exposure, foodstuff labelling, legumes, isoflavones, phytoestrogens, premenopausal women, health, endocrine disruptors

## Abstract

The study relates the present evaluation of exposure to estrogenic isoflavones of French consumers through two approaches: (1) identification of the isoflavone sources in the French food offering, (2) a consumption-survey on premenopausal women. For the foodstuff approach 150 food-items were analysed for genistein and daidzein. Additionally, 12,707 labels of processed-foods from French supermarket websites and a restaurant-supplier website were screened, and 1616 foodstuffs of interest were retained. The sources of phytoestrogens considered were soy, pea, broad bean and lupine. A price analysis was performed. A total of 270 premenopausal women from the French metropolitan territory were interviewed for their global diet habits and soy consumption and perception. In supermarkets, there were significantly less selected foodstuffs containing soy than in restaurant (11.76% vs. 25.71%, *p* < 0.01). There was significantly more soy in low price-foodstuff in supermarket (*p* < 0.01). Isoflavone levels ranged from 81 to 123,871 µg per portion of the analyzed soy containing foodstuff. Among the women inquired 46.3% claimed to have soy regularly. Isoflavone intake >45 mg/day is associated to vegan-diet (*p* < 0.01). In total, 11.9% of soy-consumers had a calculated isoflavone intake >50 mg/day. This dose can lengthen the menstrual cycles. The actual exposure to phytoestrogen is likely to have an effect in a part of the French population.

## 1. Introduction

Isoflavones are polyphenols naturally present in plants [1]. Legumes, including clover, alfalfa, soy, or lupine, are high-level sources of estrogenic isoflavones in which these compounds play essential roles. Hence, such isoflavones as genistein, daidzein or their methylated derivatives, biochanin A and formononetin attract symbiotic microorganisms involved in legume root-nodulation [2]. Legume root-nodules capture atmospheric nitrogen to fix it in the soil, and in the plant, as substrate for protein production [2]. Hence, legumes are used in crop rotation and are rich in proteins. Isoflavones can also act as phytoalexins preventing fungi infestations [3]. Considering evolution, plants with high nutritional value only managed to survive predation by developing natural defences. Therefore, pulses rich in proteins, also have several anti-nutritional components, such as anti-proteinases, phytates, oligosaccharides, hemagglutinins, saponins, or lipoxygenase [4,5,6,7]. Estrogenic isoflavones can be included in the natural arsenal to prevent predation as they can reduce the reproductive efficiency of grazing cattle [8]. The estrogenic activities of genistein and daidzein at nutritional doses have been demonstrated on many occasions, including in cell tests [9], animal toxicological studies [10,11], and in humans [12,13]. These estrogenic activities incited the OECD to advise the exclusion of diets containing soy when testing the properties of endocrine disruptors in the uterotrophic test [14]. Until recently, isoflavones were considered by a large proportion of scientists as inactive or beneficial compounds, based on the assumption that they have always been part and parcel of the Asian diet. However, it was shown recently that glycosylated isoflavones present in raw legumes are soluble in water and can be removed from soy-food by prolonged cooking, simmering or soaking in water [15]. These cooking steps were traditional in Asia and allowed inactivation and elimination of all the anti-nutritional factors [4]. However, these cooking practices are no longer found in modern soy processing [15]. This absence raises the question of the risk of consumers’ exposure to isoflavones in modern times compared to the past. This situation exists because isoflavones were not assayed in the 1930s when industrial processes were developed on soy. Conversely scientists were only able to assay isoflavones in the 1980s. Thus, most measurements have been performed on modern types of soy-food irrespective of the changes in the food manufacturing processes. Isoflavones, having estrogenic [9,10,11], anti-androgenic [16] and anti-thyroid effects [17], have been shown to interact with other chemical endocrine disruptors from the modern environment, even at low doses [18]. Isoflavones were also shown to act as endocrine disruptors on animal reproduction and behaviour [19]. Recent data correlated isoflavone presence in human biological fluids with several adverse estrogenic or anti-androgenic effects [20]. In France the last evaluation of isoflavone consumption was performed by Afssa and Afssaps in 2003 to substantiate their report on benefits and risks of phytoestrogens in the French population [21]. Therefore, consumers’ exposure to estrogenic isoflavones from modern-food requires better attention, especially with those foods that were not, until now, adequately taken into account in isoflavone databases. Here, the present-day foodstuff labelling found on French supermarket websites is explored, together with those found in foodstuffs of a restaurant supplier. The essay focused on those processed-foods in which legume-proteins could be incorporated as flakes, isolates, concentrates, or flours for either technological, economic, or nutritional purposes. The analysis without any prior notion showed that legumes are incorporated in a significant proportion of elaborated foodstuffs. A price analysis was conducted, as well as an analysis differentiating and comparing two distribution circuits: supermarket and out-of-home food-consumption. In addition, a survey was performed on 270 premenopausal women to assess their soy perception, their soy-food intake and estimate their exposure to isoflavones based on measurements performed on 150 food items containing soy or not.

## 2. Materials and Methods

We had two goals in this study. The first one was to identify food items incorporating legumes potentially containing isoflavones, which the consumer would not necessarily expect. The second one was to estimate the exposure of a category of consumer namely premenopausal women from the French metropolitan territory. For the first goal, the legumes recorded were soy, pea, broad bean, and lupine. Their contents of isoflavones as analysed in the literature are in Appendix A. The study was performed without any prior notion, considering processed-foods in which plant flours, concentrates, isolates or flakes could be easily incorporated. This included: bread products, cookies and cakes, ice cream, sauces, cooking aids, dairy products, baby foods, soups, meat-based products, fish-based products (recipes with sauce), minced meat, or fish transformed into pâté, sausages, surimi, breaded or battered meat or fish portions, kebab, pizza, canned products containing fruits or vegetables, and/or meat products. In certain cases, food products were classified as suspect, which means that their labelling mentioned that they included plant proteins or legumes or traces of legumes without any specification, but that they belong to a foodstuff category in which the pulse of interest could be incorporated. Prices were not available on the restaurant-supplier website and, therefore, the price analysis was restricted to the supermarket offering.

### 2.1. Supermarket Study 

10 French commercial retailers’ websites were analysed in the study. In some cases, as mentioned in the text, only the top 5 French retailers were taken into account. A total of 6825 foodstuffs were analysed with no prior notion about their content of legumes commonly added for technological, economic or nutritional purposes i.e., soy, pea, broad bean and lupine (Figure 1A). 

Among the different foodstuff categories analysed, some did not contain legumes while others were much too ambiguous to be retained for analysis and therefore were discarded. Many food brands are sold by several retailers. In this instance, especially for price analysis, all data concerning the same product category were collected on the same retailer web-site. Based on the analyses performed, food products were indexed according to their trademark, their price, the URL of the website where they were found, and their detailed composition as it appears to the French consumers. The number of indexed foodstuffs per category is given in Appendix A. For price analysis, only 11 food categories were selected to get enough items per group, and the analysis was performed with offers from one retailer per category collected over a week period to avoid bias due to price modifications.

### 2.2. Restaurant Supplier Study 

A restaurant supplier web-site (https://www.Mercuriale.net now indexed as https://agap-pro.com/) was analysed from April to October 2014. This restaurant supplier works with 150 food-manufacturers on one side and 1600 restaurants or cantina on the other side. Its offer is representative of the French restaurant offer. At the time of the study, this website was the only one we found that gave access to foodstuff datasheets from a panel of more than 50 different food-manufacturers. These data sheets provided comprehensive information about the foodstuffs compositions. Based on the analysis performed on the supermarket offerings, certain foodstuff categories were not analysed. These were dairy products including yoghurts, desserts, creams and cheeses, pizzas, kebabs as well as biscuits and cakes. The reason these categories were excluded is explained in the results section. A total of 5882 datasheets were found on the web-site and analysed. The datasheets of the products comparable to those found in supermarkets were indexed and analysed for their legumes contents. The number of indexed foodstuffs, as well as the categories analysed, are given in Figure 1B. For kebabs, three supplier websites were identified which were different from the main restaurant supplier web-site used in this study. Data on this foodstuff category are extracted from these websites.

### 2.3. Isoflavone Determination in Selected Foodstuffs 

Genistein and daidzein were analysed from a selection of foodstuffs based on their claimed compositions. The foodstuffs selected were either soy containing foods, soy-based foods, or legume-free foods. For each foodstuff, three samples of 1 g each of crushed fresh matter were extracted in 50 mL of stirring water. This extraction was performed for 20 min at room temperature and for 10 min at 90 °C. 500 µL of the cooled extract containing the glycosylated isoflavones were then hydrolyzed into aglycones using -glucuronidase-aryl sulfatase at 37 °C overnight, as described in [22]. The aglycones were extracted 3 times with 2.5 mL of acidified ethyl acetate that was evaporated to dryness using a speedvac. Each sample was then diluted in 500 µL of the assay buffer in a silicon-coated vial. The isoflavone analysis was performed using specific ELISAs developed by our team as described previously [23,24,25]. Haptens functionalized on different carbon atoms were synthesized as described in [23,24,25]. Specific polyclonal antibodies were obtained for each isoflavone molecule bound to bovine serum albumin and the best antibodies were retained for assay development. When cross-reactions were identified harvesting of the antibody with the reacting compound allowed to get read of unwanted reactions. Standard curve preparations and sample dilutions were performed in silicon-coated glass-vials. The ELISAs follow a competitive procedure with an immobilized competitor that is the homologous hapten bound to swine Thyroglobulin. The sensitivities vary between 0.08 ng/well and 0.4 ng/well, intra-assay variation is always below 7% and inter-assay variation is below 15%. For foodstuff the final dilution varies between 1/500 and 1/40,000. Each food item was assayed in triplicates from three extracts and on three different microtitration plates. The standard deviation reflects an inter-assay variation.

### 2.4. Survey on French Premenopausal Women 

At the end of 2017, a survey was performed on 270 premenopausal women to assess their isoflavone exposure. This survey is part of a clinical project approved under the number 2017T2-29. The questionnaire applied is given as Appendix A. The respondents were interviewed all over the French metropolitan territory with no specific geographic selection and the survey respected their anonymity. The social-economic profile of the respondent was assessed and is given in Appendix A.

### 2.5. Statistical Analysis

As this survey had a quantitative objective, it was important to estimate the margin of error associated with the different results [26]. A margin of error is the difference between the result obtained on the sample surveyed and that obtained if the entire population had been interviewed. In the case of a very large population (here, premenopausal women in France), the margin of error (*e*) depends on the size of the sample (*n*), the estimated result (*r*) and a margin coefficient (*t*), here 1.96 in the case of a 95% confidence level.
$$e=t\sqrt{r\times \frac{\left(1-r\right)}{n}}$$

According to Appendix A, if 20 % of the 270 respondents chose to answer “A”, in 95 % of cases, the result will be between 15.2 % and 24.8 % in the total target population. This error is the highest if the proportion of answer considered is at 50 % or around and then it decreases symmetrically on both sides. Thus, a margin of error is equivalent to a proportion of 30 % and 70 %. The margins of errors have been added to the observed proportions in the figures.

Then, data expressed as percentages were compared using Chi^2^ Test at a significant level of at least 5%. When the significance is higher the information is given.

## 3. Results

### 3.1. Identification of Foodstuffs from Supermarkets Containing Legumes

A complete study of more than 6600 food products was performed with no prior notion with the 10 retailers identified as commonly frequented by consumers in France (Appendix A). Legumes were searched in different categories of foodstuffs, including biscuits, baby foods, bread, breaded and battered meat or fish, cooking aids and sauces, dairy products, delicatessen, fruits in cans, legumes in cans, gluten-free products, ice-creams and desserts, bulk minced-meat fresh or frozen, minced fish (fresh or frozen), pizzas, processed food in cans, processed dishes, soy-based products, snacks, and surimi. The results obtained on eight categories of food products and classified by the type of legume added are shown in Appendix A. More comprehensive data are presented in Table 1. 

Despite the extent of the study, legumes of interest were not found in baby-foods, in dairy products including cheeses (except soy-based products and two desserts) nor in canned fruits and fruit preparations or in breaded or battered fish portions found in supermarkets (Appendix A). Foodstuff that can possibly contain pulses is bread, breaded meat, battered meat or fish and nuggets, cooking aid and sauce, delicatessen, gluten-free products, ice-creams, and desserts, legumes in cans, minced meat, processed foods in can, processed dishes, soy-based products, as well as snacks and surimi. From this table, it appears that soy is the pulse most frequently incorporated into elaborated foodstuffs containing legumes found in supermarkets. Broad bean is essentially found in bread. Pea is incorporated, as is soy, in most of the foodstuffs recorded, except for red meat-based products in which soy is preferentially used. Soy-based products were also recorded from one retailer web-site, considering that it would be representative of the global offerings. At the time of the study, 110 food-items were found in supermarket to be based on soy Appendix A gives the repartition of these 110 soy-based food items. In addition to these western-type soy-based food-products, the same retailer proposed 13 different soy-sauces and 15 Asian dishes containing soy or soy sauce. In the biscuit category, legumes and especially soy was reported in more than 80% of the products screened. However, in some cases, the soy ingredient was explicitly identified as lecithin and/or soy flour, whereas in many other cases, the nature of the soy ingredient was not mentioned. However, soy lecithin does not contain significant levels of isoflavones while, on the opposite, soy flour is very rich in isoflavone (see Appendix A). Therefore it was decided not to include foodstuffs containing either soy flour or soy lecithin without any precision in this study considering the goal of our study. Consequently, this foodstuff category was excluded from the analysis although, it may be considered as a significant source of estrogenic isoflavones. Pizza is another ambiguous foodstuff category. A comprehensive analysis of the pizza offerings was performed including 452 items. Among them, 283 items could be classified as suspect either because they mentioned plant proteins or because they mentioned traces of legumes. In addition, only two items explicitly contained soy. For this reason, pizza was not included in the analysis. However, because of its considerable consumption in France, pizza may significantly contribute to isoflavone exposure if it is made using soy flour. Analysing different supermarket websites, soy and broad beans were not found in canned form in the website offerings. However, these products can be found on store shelves. This indicates that the web-site offerings do not completely reflect what is available at the shops. Kebabs nowadays are daily consumed by more than 14% of the French consumers, and France is after Germany, the second European Kebab consumer [27]. Kebabs from both supermarket and restaurant suppliers were found to contain soy. However, the offerings in supermarkets at the time of analysis included too few items, insufficient for analysis. Therefore, a specific inquiry was performed using specialized supplier web-sites. The analysis is detailed with the restaurant supplier analysis.

### 3.2. Respective Proportion of Each Selected Legume in the Main Foodstuff Categories 

After having identified the categories of foods containing legumes, it seemed relevant to determine the proportion of legume-containing foodstuffs in each category. For this analysis, all of the offerings of one of the top 5 retailers were fully examined for each food category. The foodstuff categories containing pulses were then analysed according to the presence of soy, lupine, broad bean, or pea. The respective proportions of food containing the four added legumes are presented in Table 2. This table gives data obtained in supermarket besides those obtained analysing the offer of the restaurant supplier. Table 2 also gives the percentages of soy containing items in each category together with the significance of the difference between the two offers. The results of this comparison will be discussed later.

### 3.3. Study by Price 

An initial analysis of the main food categories was performed classifying the food-products by their prices. This analysis was performed on eleven food categories presenting sufficient items, including breaded meat, breaded fish, burgers, delicatessen, fish steak, meatballs, minced meat in bulk, minced meat in portion, nuggets, stuffed vegetables, and surimi. In this initial analysis, all foodstuffs were treated together and dispatched into 7 price categories from 2.5 € to > 17.51 € per Kg (i.e., 2.50€–5.00€, 5.01€–7.50€, 7.51€–10.00€, 10.01€–12.50€, 12.51€–15.00€, 15.01€–17.50€, > 17.51€). This analysis did not show any significant differences between the proportions of the different legumes in each price category. A second analysis performed within each foodstuff category did not manage to show any significant findings because, generally, the number of food items was low, and the offerings were heterogeneous. Finally, an analysis was performed defining price-quintiles for each food category. The quintiles were different depending on the foodstuffs being considered (Appendix A). The numbers of products containing soy, pea, no legumes or suspects were considered for each price-quintile. The number of products claiming to contain legumes or not (and percentages) per price-category is given in Table 3.

Considering each price quintile individually the table shows that in the two lowest price-quintiles, the percentage of foodstuff with no claimed legumes is approximately 50% (52 and 49% for price 1 and 2 respectively). At the same time, in the three upper price-quintiles, the percentage of foodstuff devoid of legumes ranges from 68% to 77%. When the two lowest price-quintiles are considered together, there is a significant difference (*p* < 0.01) between the proportion of pulse containing items in the two lowest price quintiles and the three uppers. Legumes are more frequent in the lowest quintiles. When looking at soy content, it appears that the proportion of soy containing foodstuff is higher in the two lowest price quintiles (*p* < 0.01) than in the three uppers.

### 3.4. Theoretical Exposure to Isoflavones from Canned Vegetable Preparations 

Because isoflavones are known to be present in small amounts in traditional French pulses, a theoretical calculation of the resulting content was attempted to analysing the offerings of canned legumes from a specific retailer. The expected isoflavone concentrations expressed in aglycone equivalent in the legumes of interest, come from the literature (see references in Appendix A). The number of food-products was considered an indication of demand by the consumers since the offerings should be in line with consumer expectations for the retailer to maintain a good financial balance. Appendix A shows the expected amount of isoflavones (aglycone equivalent) in 100 g of each preparation-type found on a supermarket website and Appendix A shows the proportion of each canned legume or legume-preparation in the retailer’s offer. Globally, the isoflavone content is several tens of µg per 100 g except for mungo beans and mixtures containing peas. In these cases, the isoflavone amounts reach 1 to 2 mg per 100 g of fresh weight. These isoflavone intakes are traditional in France.

### 3.5. Assays Performed on Some Foodstuffs from the Supermarket Offerings

Several food items corresponding to different categories were assayed for the main isoflavones, i.e., genistein and daidzein in aglycone equivalent (Appendix A). The data are expressed in µg per g or per portion since these are the conventional units for other endocrine disruptors. The food assayed included foodstuffs based on soy-juice, Asian types soy-based dishes, prepared dishes based on soy, soy products, health products based on soy, foodstuffs with hidden soy. As indicated in Appendix A, soy-based products, with the exception of soy lecithin, contain large amounts of isoflavones. Toasted soy grain used as appetisers, is by far the food item containing the largest proportion of isoflavones, although the portion size is usually rather small. Several tofu preparations were assayed. All were found in shops belonging to trade groups and were industrial preparations. The whey collected when opening the tofu packaging was more concentrated in isoflavones than the tofu itself. This shows, that isoflavones as glycosylated substances are soluble in water [15]. Soy sauce contains isoflavones but considering the way it is used its contribution to isoflavone intake remains low. The food items in which soy was incorporated as an ingredient for technological, economic or nutritional reasons contained various amounts of isoflavones from 0.08 mg per portion up to nearly 20 mg per portion. Some food-items, claiming soy as an ingredient, were found not to contain isoflavones, and this will be discussed later. The concentrations of isoflavones indicated in this study should not be considered as granted since variations can occur from one batch to another and from one trade-mark to another. To illustrate the variation between trade-marks and batches, 11 soy-juice trade-marks were analysed for their content in genistein and daidzein. The results are given as Appendix A and show isoflavone concentrations in aglycone equivalent varying from 9.31 to 92.89 mg for a mug (330 mL). Analyses showed that isoflavones were not present in the 10 food-items claiming not to contain soy. Among the 15 foodstuffs assayed claiming to contain soy, four exhibited too low isoflavone levels to be considered as being adequately labelled. These foodstuffs were one meat-ball batch claiming to contain 15% soy protein, one burger trade-mark claiming to contain 12% soy protein in the meat-portion, one breaded meat batch claiming to contain 10% soy protein. Small sausages for parties claiming to contain soy were also found to have undetectable isoflavone levels.

### 3.6. Analysis of the Restaurant-Supplier Website

The restaurant-supplier website was recently modified and the access to the foodstuff datasheets used for this study is no longer possible. The complete list of the indexed products is still available from the database built for this study and the datasheets, having been indexed, are available upon request. The global study made without any prior notion using the supermarket web-sites allowed the food categories containing added legumes to be undoubtedly identified (See Appendix A). These categories were compared for the two distribution circuits. Figure 2 shows the percentage of foodstuffs containing legumes in each category, except the minced-fish portion. 

For some categories, including breaded fish, delicatessen, and minced meat portions, the numbers of references were large (109, 139, and 69 respectively). For the other food-items, the offerings were smaller with the number of foodstuffs ranging from 5 (bulk minced-meat) to 28 (breaded meat). In some cases, 100% of the offering included products that claimed or were suspected to contain pulses. These were stuffed vegetables, meatballs, and burgers. Three restaurant-supplier websites specializing in kebab products were analysed. It was found that among 12 products, half of them claimed to contain soy at the time of the analysis. No prices were given for these preparations. No specific study was performed on these products, although kebabs are becoming ever increasingly popular with French consumers. Therefore, they probably contribute significantly to the still underestimated intake of isoflavones.

### 3.7. Comparison of the Two Distribution Circuits

The comparison was achieved for 9 foodstuff categories to get enough items for statistical analysis (Figure 1B). For the restaurant supplier, the offerings included 354 food-items. The corresponding food-items were selected from the top-5 French retailers but for each type of foodstuff, only one supplier offering randomly defined was considered. An exception was made for Surimi, for which the offerings being very homogeneous were analysed for various retailers’ websites. Table 2 shows the number of products per category and the percentage of products containing soy or any legume or suspect. The table shows that the percentage of soy-containing products was significantly higher in restaurants (*p* < 0.01). For legumes the overall difference is not significant but the Surimi category is likely to contain more frequently legumes in restaurant (*p* < 0.01). 

### 3.8. Consumption Survey and Isoflavone Exposure

The consumption questionnaire is given as Appendix A. It contains four main issues: (1) a global approach of food consumption, (2) an inquiry on soy consumption, (3) an inquiry of soy perception, (4) a social-demography approach. Only the data obtained on Soy consumption and potential isoflavone exposure will be illustrated here. Other results will only be commented on. 

(1) In the global approach of food consumption, 52.6% and 14.4% of the interviewed women claimed to pay attention and great attention respectively to their diet. 7% claimed to be vegetarian and 3.7% claimed to be intolerant to several food components. 88.5% of the inquired women buy their food in supermarkets. However, they also use other distribution circuits, and the second most often cited is the city market (25.9%). Among the interviewed women 73.7% never have their food in canteens or have it less than once a week, and 75.2% never have their food in fast-foods or food-trucks or have it less than once a week. Among foodstuffs that are likely to bring unexpected soy the most popular item is minced beef portion and hamburger for which 56.7% of the women inquired claimed consumption at least once or twice a month. Based on Appendix A a calculation of isoflavone exposure through a casual diet containing hidden soy can be proposed. The data of exposure with their confidence intervals are presented in Table 4. 

According to this calculation, the mean intake in the population inquired is 2.08 mg/day when the median is 1.06 mg/day. The population inquired is roughly distributed into thirds regarding its attention paid to labels. A third is vigilant on all foodstuffs, a third only on several products and a third is not vigilant at all. Among those who claimed to pay attention to labels the two first items that are considered seem to be the factsheet and composition table and the geographic origin. For 75.1% presence of soy among other ingredients on a label has no impact on choice. 

(2) Soy and isoflavone consumptions are given in Figure 3.

As seen in Figure 3A the percentage of inquired women that declare having soy regularly is 46.3% CI_5%_ (40.4%–52.2%). 13% of the non-consumers had soy in the past and stopped mainly for taste and health issues (5 stopped for health reasons over the 19 who stopped). According to Figure 3B, most consumers started having soy recently since 49.6% started during the past three years (from 2015 to 2017). The most frequently consumed soy-based foodstuffs are soy-based dairy products (i.e., yoghurt, dessert cream, cheese, and soy-milk). Among the 125 consumers 47% had dessert cream and yoghurt at least once a week. Similarly 38% had soy-based drinks at least once a week. Besides soy-based cookies, pancakes and cakes are consumed at least once a week by 42% of the inquired population. According to Figure 3C the most frequent category among soy consumers is the one having 1 to 3 soy-food portions a week. The highest consumers can have >25 soy portions a week. From the inquired panel it appears that the first reason to eat soy is taste (56.8%) and the second is to replace animal proteins (33.6%). For 30% of the panel soy is better for health. Most consumers had no favourite trademarks (68.8%). From a table of isoflavone concentrations in soy-based foodstuff and the consumption frequency an index of plausible isoflavone exposure was built as seen in Appendix A. From these data it was possible to build Figure 3D. From this figure it appears that nearly 12%, 11.76%–CI_5%_ (8.0%–15.7%) of the population has a daily isoflavone (eq aglycone) intake over 50 mg/day. This dosage is known to lengthen the menstrual cycle of premenopausal women by two days. In most cases (76%) soy-based products are restricted to adult consumption (inquired person and partner). Despite the last advice by Anses, the French Food Safety Agency in its EATi report published in 2016 [28], 1.6% of children below 3 years old are concerned with soy consumption. Table 5. shows the repartition by age category of high isoflavone consumers.

It is seen from this table that some children can be exposed to large amounts of isoflavone especially considering their body-weight. This proportion, however, seems low. Only 2.6% of the consumers exposed to isoflavone doses (eq aglycone) >45 mg/day are children below 3 years old and only 5.3% of the consumers that are exposed to isoflavone doses (eq aglycone) >45mg/day are children over 3 and below 12 years old. When analysing further, it is seen that an isoflavone intake >45 mg/day (eq aglycone) is significantly associated with the vegetarian status (*p* < 0.01). Only 20 persons declared to be vegetarian in the panel inquired, that is 7.40% of the panel interviewed. Hence, there is also a significant association (*p* < 0.01) between isoflavone intake >45 mg/day (eq aglycone) and organic shop purchasing. Finally, there is a significant (*p* = 0.04) link between an isoflavone intake >45 mg/day (eq aglycone) and everyday sports practice. In parallel there is no significant association between family incomes and isoflavone intake >45 mg/day nor between Hard discount purchasing and isoflavone intake >45 mg/day (eq aglycone).

(3) Soy food perception. 67.4% of the interviewed women spontaneously associate soybean with a healthy and balanced diet. 65.2% of the inquired women reported that they were never advised having soy. Among those who answered yes to the previous question, 87.2% said that they were advised by a relative or by media. Health practitioners were only cited in a few cases and often it was the family doctor (8.2%). At this stage of the survey, among the 270 women that were interviewed 28.1% considered that soy is overall beneficial, 30.3% considered it is neutral and 8.5% considered it is adverse and 33 % have no opinion. The main top 3 issues why soy may be adverse are for babies, for hormonal cycles and for fertility. Surprisingly, nearly 22% of the persons interviewed said that soy is good for allergies. Finally, 21.9% of inquired women think that plant-based infant formulas are better than animal based infant formulas, and 75.2% of the panel considered not to be sufficiently informed on the health effects of soy.

(4) Socio-economic profile of the respondents. The results are given in Appendix A together with the National data collected by Insee (National Institute of Statistics) in 2017–2018. Globally speaking the population is younger than the global population of premenopausal women. Considering the family composition, the sample seems to be equivalent to the national means. There are fewer people with children in our panel reflecting the young age of the cohort. The monthly incomes are shifted to an upper level in our panel.

## 4. Discussion

### 4.1. Presence of Pulses in Processed Food

#### 4.1.1. Under-Estimation of Pulses and Isoflavone Exposure 

Isoflavone amounts in the µg range per 100g can be found in canned legumes traditionally eaten in France (pea, beans, broad beans, chickpeas). Besides, it was shown here that a significant intake can be identified from casual processed food containing soy as an ingredient. The median isoflavone exposure (aglycone equivalent) was found to be 1.06 mg/day (mean being 2.08 mg/day aglycone equivalent). This calculation is far to be precise since it may have over-evaluated the amount of isoflavone in the foodstuff considered on one side and it did not take into account other foodstuffs that are potential isoflavone providers, such as legumes, biscuits and cakes, pizzas, or kebabs for instance. Therefore, isoflavones may not only be present in vegetable-based foodstuffs but also in processed food. Here, it was found that legumes were present in 41% of processed foodstuffs found in supermarkets and soy in 11.76%. Considering soy, lupine, pea, and broad bean as ingredients, soy is by far the pulse with the highest content of isoflavone. However, evaluating isoflavone exposure on the individual basis of soy-based product intake only [21], not taking into account processed food containing pulse as ingredients or based on other pulses, is likely to underestimate the exposure to isoflavones [29]. In France, as mentioned previously the exposure to genistein and daidzein was assessed in 2003 and was estimated at 0.151 mg/day because only soy-based foodstuffs were considered and they were found to be eaten by only 0.1% of the French population [21]. In parallel, the isoflavone intake estimated from soy-based foodstuffs was found to be 0.025 mg/day. In the meantime, the European daily isoflavone intake was found to be approximately 1 mg [30]. However, more recently, it was reported a mean phytoestrogen exposure of 18 mg/day for British post-menopausal women, when lignan and phytoestrogen-based food-supplements consumption were also considered [31]. In parallel and in the same period, the USA daily exposure to estrogenic isoflavones was reported to be approximately 2.7 mg or less [32,33]. The present study support previous findings [34], showing that the proportion of soy consumers dramatically increased in France recently (41% in 2016 vs. 0.1% in 2003). It also shows that up to tens of milligrams of isoflavones can be found in a portion of processed food containing legumes as ingredients (Appendix A) and that biscuits, cakes and pizzas have too ambiguous labelling to foreseen their contribution to phytoestrogen exposure. The study also shows that the food-items considered are widely consumed by the French population. All factors combined should significantly increase the global estimation of the isoflavone intake by the French population. If, as in France, USA minced-meat portions, burgers, meatballs, chilli, kebabs, and bread, as well as cakes, contain added soy proteins, then the exposure to isoflavones could be much higher than what was previously described. Although no data are yet available in France about the isoflavone content in consumers’ urine, such a ubiquitous distribution has been noted from data collected in Germany [35], in Israel [36] and in the USA [37] which showed ubiquitous presence of isoflavones in human urine samples. Nevertheless, the isoflavone exposure in European countries is lower than what is currently observed in Asian countries where people eat industrial soy products. The long term consequences on reproductive physiology of such practices still have to be fully assessed [38,39]. The present study should also prompt epidemiologists to re-evaluate Western consumers’ exposure to isoflavones. This should help in explaining some discrepancies between isoflavone intake evaluation and their corresponding plasma levels even when the intake is rigorously evaluated [31]. Moreover, as far as epidemiological studies are considered, the underestimation of isoflavone intake, when it lies in the mg range, can possibly induce bias in case-control studies and probably reduces the power of human observational studies. Since the same isoflavone intake was shown to induce highly variable isoflavone plasma levels (factor 10 of inter-individual variation) [40], this impact on clinical observations should be considered. In this context, if the isoflavone concentrations are not monitored in the volunteers’ biological fluids, then this can lead to misinterpretations of the results.

#### 4.1.2. The Differences between the Distribution Circuits

The restaurant-supplier organizing the distribution of foodstuffs from 150 food-manufacturers to more than 1600 restaurants or restaurant clusters is representative of the out-of-home catering. It was shown here that it offered about twice as many products containing soy than did the supermarkets. Our study showed that 25.71% of processed food for restaurant and canteens contained soy. The difference with supermarket is highly significant (*p* < 0.01). As an example, breaded-fish brands containing soy were found on the restaurant-supplier web-site only. This can be explained because restaurant-suppliers of school canteens must guarantee fishbone-free products. Therefore, the fish is ground and legumes are added to ensure the texture of the final product before being breaded. This would indicate that the exposure to legumes and isoflavones could be different depending on whether or not the consumer eats processed-food at home. In addition, pulses were found in food-products particularly popular among youngsters. If the exposure is calculated on a body-weight basis, this increases the risk of a final physiological effect as is discussed below. The data presented here should be taken into account when assembling new databases used to evaluate isoflavone intake.

#### 4.1.3. The Price Analysis

The price survey shows that soy and pea are significantly most often present in low-cost food items (*p* < 0.01). However, the incorporation of legumes in processed-foodstuffs is not only directed by economic reasons. If so, the global survey of the prices of foodstuff-containing-legumes would have shown a price effect. In addition, there was no difference between classical retailers and hard discounters, considering the presence of legumes in processed-food sold under supplier trade-marks. 

### 4.2. Analysis of the Premenopausal Women Survey

The data presented here had never been recorded before and they show several points. First, the mean exposure via “hidden” soy in this specific category of consumers is about 1 mg/day aglycone equivalent because most of the premenopausal women inquired (>73%) do not frequent canteens or fast-foods and food-trucks and because their consumption of processed dishes potentially containing soy is limited. However, soy consumers represent nowadays almost 50% of the premenopausal women population in Metropolitan France and 12% of the inquired women had soy several times a day exposing them to isoflavone doses over 50 mg/day aglycone equivalent. As mentioned earlier it is known that doses of 40–50mg/day aglycone equivalent can lengthen the menstrual cycle of premenopausal women by two days [12,41]. This is why such isoflavone level thresholds were chosen for association analysis. The level was chosen to get samples big enough for Chi^2^ Test analysis. The results show that vegetarian women are more likely to reach active isoflavone intake via their food. People with healthy behaviour (organic shop purchasing and everyday sport practice) are more likely to be overexposed to isoflavones. Finally a small proportion of children can be exposed to large dose of isoflavones when adjusted on their body-weight. Soy perception seems to be unclear in the consumers’ mind. On the one side 67.4% of our panel associate soy to a healthy and balanced diet, on the other side only 28.1% associate it to beneficial effects. Hormone-like effects seem to have been perceived by a small part of the population. Conversely nearly 22% of the population would use soy in case of allergy. Such result can be explained because soy-based infant formulas were used in the past in France [42] to face lactose intolerance in babies and the public does not make difference between allergies and intolerances.

### 4.3. Consequences of Plant Protein Incorporation in Processed Food

#### 4.3.1. Technological and Nutritional Arguments

Legumes are incorporated into processed food for technological reasons [43]. Scientific data are available showing the emulsifying properties of plant proteins, as well as their whitening properties [44,45]. In addition, plant proteins present interesting amino-acid profiles as previously indicated [46]. Previous studies have shown that soy, pea and lupine proteins exhibit similar properties. In some cases, soy appears to be better adapted to human nutrition and in other cases, pea has greater nutritional qualities [46]. The discrepancies between studies probably come from differences in varieties or genetically improved qualities with the course of time [47]. Nevertheless, the assays performed in this study suggest that soy could be replaced by another plant-protein. This could be suspected because of unfitted proportions of isoflavones in some processed-foodstuffs claiming containing soy. This is explained because, for the same trademark, some food-items are not all produced in the same place of the French territory. Previously, each production unit could choose its own ingredients to prepare its processed-dish. This was done by adjusting supplies according to availability and costs, providing that the safety of the final product was ensured. This was allowed even under a common label. Then, it was found that, for some nuggets, battered or breaded meat, soy was declared as an ingredient but isoflavones could be hardly detected. This indicated that soy was probably replaced by another legume (most likely pea) and demonstrated the feasibility of the substitution. In addition, soy is officially recognized as an allergen whereas pea is not. If the replacement did not induce any technological or economic changes and reduced the risk of allergy, it seemed to be considered as an acceptable practice. However, the consumer protection law was enacted one year after our food analyses and this kind of practice is probably reduced nowadays [48].

#### 4.3.2. Allergenic Effects

Soy and Lupine, being allergenic ingredients, are more extensively present in the French food offerings than expected. Then, the consumer protection law specifies to mention soy and lupine in bold on food labels [48]. Hence, health consequences should be considered. By the way, there has been an increase of acute allergic reactions to food in developed countries in recent years. Food reactions are not always due to allergies and many different reactions are known to occur following dietary exposures [49]. In soy, the allergenic effect is mainly related to beta-conglycinin (Gly m5) and glycinin (Gly m6), two storage proteins [50]. These two proteins are stable when heated and can induce chronic and acute reactions. They can also act via interactions with the peanut allergens, vicilin (Ara h1) and glycinin (Ara h3) [51]. Gly m4 (PR-10 protein, Bet v1 homologue, 17 kD) and oleosin (24 kD) can also be responsible for severe allergic reactions after soy-juice intake as demonstrated in two case-reports [52]. Gly m4 is destroyed by heating and during digestion, and its effect is usually associated with oral reactions. Lupine also contains such allergenic proteins as gamma-conglutin (Lup a), which was shown to induce cross-reactivity with allergenic proteins in peanuts [53]. Pizzas, as previously mentioned in this text, have ambiguous labelling. As such, pizzas are commonly reported as allergenic foods. In some cases reported [54], the allergen was found to be soy. Recently, in France, the allergo-vigilance system has identified an increasing problem with soy allergy due to the increasing use of this pulse in processed food [55]. The manufacturing process seems to influence soy allergenicity [56]. 

#### 4.3.3. Soy Phytoestrogen Content and Health Effects

As mentioned in Appendix A, the isoflavone content of the 4 legumes retained in this study is not equivalent. Soy is by far the largest provider of estrogenic isoflavones. In soy, these compounds that are mainly genistein and daidzein, are known as phytoestrogens. Their estrogenic activities were demonstrated in vitro in animals and in humans [10,11,12,14,57]. Estrogens are involved in many physiological pathways from reproduction, cell proliferation and immunity where their effects can be deleterious, to bone health, memory, or climacteric symptoms where they can be of help. Estrogens can, therefore, be both beneficial and deleterious according to the person’s physiological status. According to previous studies, in premenopausal women, the isoflavone active dose ranges between 40 and 50 mg/day [12,41]. At that doses in aglycone equivalent and higher [58,59], genistein and daidzein from soy lengthen the menstrual cycle by acting at the pituitary level on GnRH, FSH and LH secretion [12,41,60]. In some studies, consequences of this central disruption on estrogen and thyroid hormone synthesis were observed [58,59]. Disruption of FSH and LH in men is known to reduce sperm count and quality. Nowadays, five different studies i.e., two American [37,61], one Japanese [62], and two Chinese [63,64] reported correlations between increased isoflavones in biological fluids and reduced sperm quality and quantity. In these studies, interactions of soy-isoflavones with endogenous estrogens and/or xenoestrogens could never be refuted. In addition, recent work showed a positive correlation between genistein plasma levels in women and the expression of genes involved in estrogen-dependent breast-tumour growth [65]. The effect of soy isoflavones in food supplements was also shown on triple-negative breast tumours of women with family risks of breast cancer, probably due to the activation of tumour cell proliferation by phytoestrogens via the GPR30 receptor [66]. Hence, the available data indicate a protective effect of isoflavones at the initiation step of estrogen-dependent breast cancers and, on the opposite, a deleterious effect of the same compounds at proliferation stage of the corresponding tumours [67]. Therefore, it is not surprising to see a difference of phytoestrogen effect on breast cancer during the first year of phytoestrogen-based food-supplements intake and after [68]. During the first year a tumour that is already present can be revealed while after one year, the healthy breast that did not reveal a tumour is protected. Moreover, interactions of soy with thyroid function in human are known from the 1960s especially in patients with pre-existing hypothyroidism and those under levothyroxine [69,70,71,72]. More recently, isoflavones were demonstrated to be responsible for this interaction in humans with mild hypothyroid status [17] and to have transient increasing effect on rT3 plasma concentrations in healthy postmenopausal women [73]. Finally, environmental estrogens are currently considered as endocrine disruptors when they reach the consumers in an uncontrolled fashion. They are increasingly present in our environment and as shown by previous authors, isoflavones with estrogenic properties that are now ubiquitous in industrialized countries can act synergistically at low doses with other endocrine disruptors such as pesticides [74,75] or wrapping agents [19]. Confirming all these clinical data, in 2008, the National Toxicology Program in the USA showed reprotoxic [10] and carcinotoxic effects [11] of genistein from soybean. Such studies can provide Reference Toxic Value. This should lead to the reduction of isoflavone in soy-based product probably going back to ancestral cooking practices in water [13,15,76].

### 4.4. Limitation of the Study

This analysis is one of the first undertaken in Europe to assess the presence of pulses containing isoflavones in the consumers’ dietary environment. It is also the first study comparing data from two distinct supplier-circuits and to complete the approach by a survey on a panel of respondents. However, in considering these approaches, several limits should be noted. (1) The label analysis was performed from web-sites which were found to not exactly reflect what could be found on the shop’s shelves. Moreover, new food items were not always found on the web-sites. (2) Although more than 12,000 processed food-items were screened in total, approximately 9 to 10% only seemed to contain legumes (with the exception of cakes, cookies, pizzas and kebabs), even if this proportion could vary from one foodstuff category to another. (3) Only one restaurant-supplier representing 150 food manufacturers and 1600 restaurants of restaurant clusters was considered since it was the only one for which the complete food-item datasheets were available at the time of the study. (4) The study did not consider raw products (fruits, vegetables, raw pasta or non-transformed meat), which represent a large component of a supermarket’s offerings. (5) The study is based on the information mentioned by the manufacturers on the label of their products. However, it was noticed that this information is not always precise-enough to avoid ambiguity. The nature of the plant protein preparation (flour, protein isolates, protein concentrates, and flakes) incorporated in the food items, was generally not indicated although this can make large differences in the isoflavone content (Appendix A) [77]. The ambiguity of the labels, especially for cakes, cookies, or pizzas, has already been noted for American foodstuffs [78]. In some cases, the assays demonstrated that the information on the label was false because isoflavones were found in unfitted proportions. Whatever the distribution circuit, the analysis showed that approximately 10% of the products claiming legume contents or traces were considered as suspect. This proportion was too high (approximately 80%) in biscuits and cakes and about 50% in pizzas to allow any relevant analysis. Nevertheless, because these food items are commonly chosen by the consumers, this analysis has most likely underestimated the plausible sources of legumes and therefore of isoflavones for the consumers. (6) This study tended to bring data on isoflavone exposure through a survey directed specifically toward premenopausal women and through isoflavone assays in a rather large quantity of food items (150 in total). The number of isoflavones can vary from one soy-based foodstuff to another and not all casual diet contains the same amount of soy and of isoflavones. Therefore, the mean level of isoflavones considered to calculate the exposure can be debated. As mentioned earlier, if there may be an over-estimation on one side, there may also be under-estimation on the other side, because not all isoflavone-containing foods have been considered (this includes traditional French legumes, biscuits and cakes, pizzas, kebabs containing soy as an ingredients etc...). The percentage of premenopausal women having identified soy is in accordance with a study previously published in France in the same period considering a global soy consumption of 41% of the adult population (men and women included) [34]. (7) Although the panel does not completely reflect the premenopausal French population (see socioeconomic data Appendix A) it brings new results but it should not be extrapolated to other population groups especially children or young male adults for which the consumption habits are most probably different.

## 5. Conclusions

Legumes containing a significant amount of estrogenic isoflavones were found as ingredients in many modern processed-foodstuffs. This situation is likely to evolve rapidly, considering the creativity of the food industry. These modern foodstuffs include meat and fish-based products which are not those taken into account in conventional isoflavone databases. For some popular food-items, the percentage of soy-containing-food can range from 30 to 100%. Over a panel of processed food items found in France there is twice as much chance to have soy in restaurant than in supermarket. Some popular processed-foods such as nuggets, burgers, meatballs, kebabs, pizzas or breaded fish can contain soy with significant amounts of estrogenic isoflavones. The reasons legumes are incorporated into processed foodstuffs are all economic, nutritional or technological. Low-cost foodstuffs significantly contain soy more frequently. Nearly 50% of premenopausal women declared to have soy-based products regularly in the survey performed in fall 2017. A significant proportion of women who are more likely to be vegetarian, and to have healthy behaviour, have an isoflavone intake over 50 mg/day (aglycone equivalent). At this dose the menstrual cycles are lengthened. Children isoflavone intake is usually reduced to a few percentages of the population but considering their low body-weight the relative exposure can be high. Estrogenic isoflavones reaching consumers unintentionally can be considered as endocrine-disrupting agents and may act in synergy with other synthetic endocrine disruptors. Their toxic effects were previously shown on reproduction and on mammary and pituitary carcinoma [10,11]. Reference Toxic Value should be defined according to these data. Reducing this exposure could make sense and may be achieved by substituting other plant protein sources for high-isoflavone-containing legumes as long as the technological, economic, or nutritional issues are respected. Another approach could be to modify soy manufacturing processes known to influence isoflavone concentrations [15].

## Figures and Tables

**Figure 1 nutrients-11-02308-f001:**
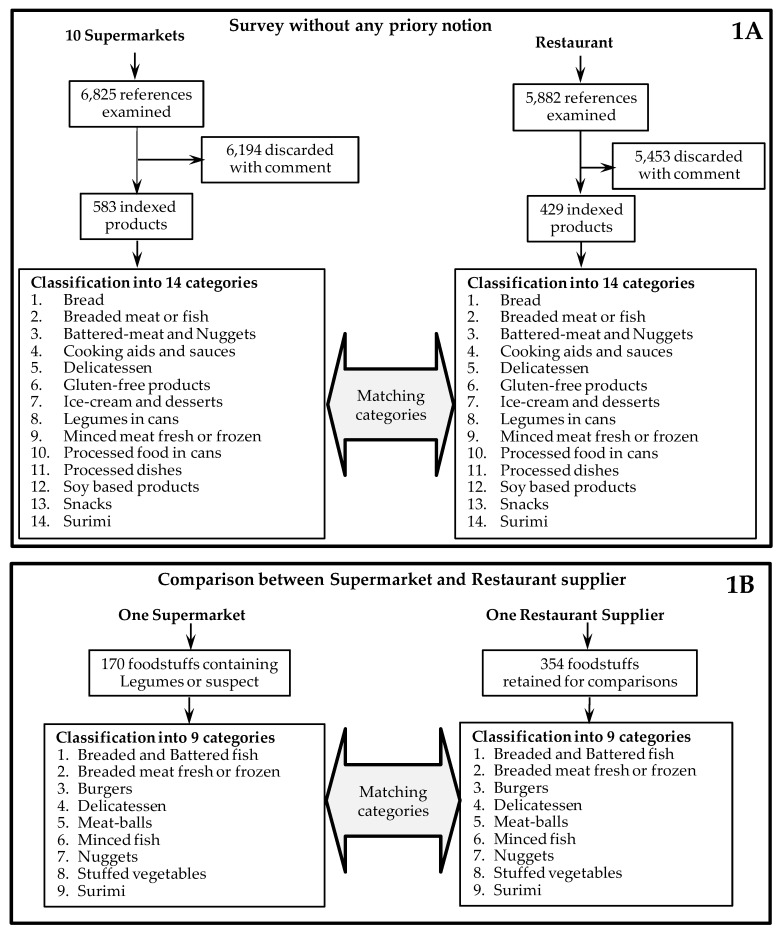
Diagrammatic representation of the food-item sorting considered in the different studies. (**1A**) Sorting applied to the global study of supermarkets and of the restaurant-supplier offerings. (**1B**) Sorting applied to the comparison between the two distribution circuits.

**Figure 2 nutrients-11-02308-f002:**
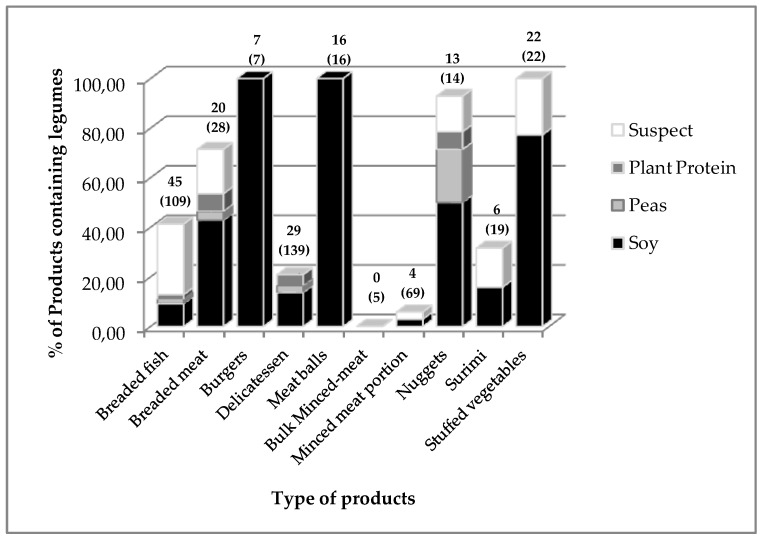
Percentage of food-items containing legumes in the different food-categories proposed by the restaurant-supplier. Figures indicate the number of food-items and, between brackets, are the numbers of total food-products found within the restaurant-supplier offering.

**Figure 3 nutrients-11-02308-f003:**
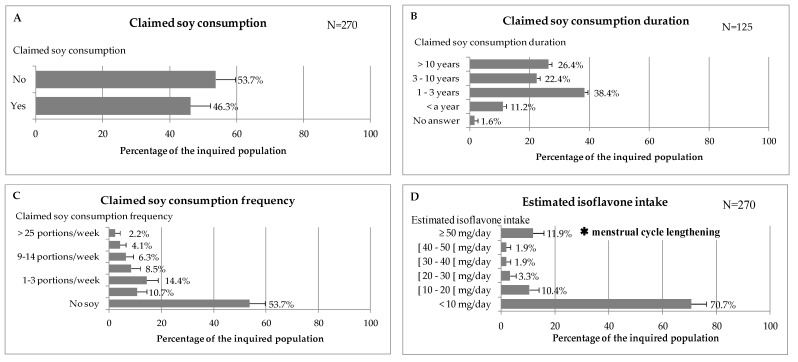
Main results of the survey performed on 270 premenopausal French women on their soy consumption. (**A**) Percentage of premenopausal women consuming soy-based products or not. (**B**) Duration of soy-based product consumption over women claiming regular soy intake. (**C**) Frequency of soy intake from all the inquired population (*N* = 270). (**D**) Estimated isoflavone intake based on the isoflavone foodstuff estimation (Appendix A). Data are percentages and error bars are confidence intervals calculated with a risk error of 5%.

**Table 1 nutrients-11-02308-t001:** Repartition of the pulses of interest in selected foodstuff categories from supermarkets.

Food Categories	Number of Foodstuffs						Total of Indexed Foodstuff	Products Claiming Containing Pulse
	Soy	Pea	Lupine	Broad Bean	Suspects	Traces		
1. Bread	11	-	-	37	3	1	52	48
2. Breaded meat or fish	12	14	-	-	3	-	29	26
3. Battered-meat and Nuggets	5	10	-	-	6	-	23	15
4. Cooking aids and sauces	22	-	-	1	9	-	32	23
5. Delicatessen	11	10	-	-	2	-	26	21
6. Gluten-free products	13	6	3	-	3	-	27	22
7. Ice-cream and desserts	7	3	-	-	6	-	16	10
8. Legumes in cans	0	33	4	-	-	-	54	37
9. Minced meat fresh or frozen	52	7	-	-	5	-	65	59
10. Processed foods in can	13	12	1	-	-	-	26	26
11. Processed dishes	23	18	-	1	15	-	57	42
12. Soy based products	138	-	-	-	-	-	138	138
13. Snacks	12	9	-	1	10	-	32	22
14. Surimi	4	-	-	-	2	-	6	4
Total	323	122	8	40	64	1	583	493

**Table 2 nutrients-11-02308-t002:** Comparison between the offers of the processed food claiming to contain pulses in the two distribution circuits.

Product Category	Supermarkets					Restaurants					Significance Products with Legumes vs. no Legumes	Significance Products with Soy vs. no Soy
	Products per Category	Products with Legumes or Suspect	% with Legumes or Suspect	Products with Soy	% with Soy	Products per Category	Products with Legumes or Suspect	% with Legumes or Suspect	Products with Soy	% with Soy		
1. Breaded and Battered fish	14	1	7%	0	0%	109	45	41%	10	9%	*na*	*na*
2. Breaded meat fresh or frozen	18	15	83%	4	22%	28	20	71%	12	43%	*ns*	*na*
3. Burgers fresh or frozen	7	4	57%	2	29%	7	7	100%	7	100%	*na*	*na*
4. Delicatessen	49	9	18%	0	0%	139	29	21%	19	14%	*ns*	*na*
6. Meatballs fresh or frozen	10	10	100%	5	50%	16	16	100%	16	100%	*na*	*na*
7. Nuggets fresh or frozen	24	19	19%	2	8%	14	13	93%	7	50%	*na*	*na*
8. Stuffed vegetables fresh or frozen	5	4	80%	3	60%	22	22	100%	17	77%	*na*	*na*
9. Surimi *	43	7	16%	4	9%	19	6	32%	3	16%	*p* < 0.01	*na*
Total number of products (%)	170	69	41%	20	11.76%	354	158	45%	91	25.71%	*ns*	*p* < 0.01

*na*: not applicable, *ns*: not significant, * Surimi from different supermarkets to cover the entire offer.

**Table 3 nutrients-11-02308-t003:** Price analysis per foodstuff categories.

Price-Quintiles for All Product Categories*	Pea	Soy	Plant Protein + Suspect	Total Legumes	No Legumes	Total	Significance Legume vs. no Legumes
Price 1	7 (15%)	11 (23%)	4 (9%)	22 (48%)	24 (52%)	46	*na*
Price 2	14 (27%)	10 (19%)	3 (6%)	27 (51%)	26 (49%)	53	
Price 3	6 (12%)	1 (2%)	4 (8%)	11 (23%)	37 (77%)	48	
Price 4	6 (13%)	2 (4.3%)	6 (13%)	14 (32%)	30 (68%)	44	
Price 5	1 (4%)	3 (12%)	4 (16%)	8 (32%)	17 (68%)	25	
Price 1 + 2	21 (14%)	21 (14%)	7 (5%)	49 (33%)	50 (34%)	99	*p* < 0.01
Price 3 + 4 + 5	13 (9%)	6 (4%)	14 (9%)	33 (22%)	84 (56%)	117	
							**Significance soy vs. no legumes**
Number by category	34	27	21	82	134	216	*p* < 0.01
Price 1 + 2	21 (62%)	21 (78%)	7 (33%)	49 (60%)	50 (37%)	99	
Price 3 + 4 + 5	13 (38%)	6 (22%)	14 (67%)	33 (40%)	84 (63%)	117	

* The quintiles are given in Appendix A, *na*: not analysed.

**Table 4 nutrients-11-02308-t004:** Distribution of isoflavone exposure in French premenopausal women through a casual diet containing soy as an ingredient.

Classes of Isoflavone Exposure	Number of Subjects	Confidence Intervals
< 0.5 mg /day	84	25.6% < 31.1 < 36.6%
(0.5–1) mg/day	48	13.2% < 17.8 < 22.3%
(1–2) mg/day	37	9.6% < 13.7 < 17.8%
(2–3) mg/day	31	7.7% < 11.5 < 15.3%
(3–5) mg/day	42	11.2% < 15.6 < 19.9%
(5–8) mg/day	21	4.6% < 7.8 < 11.0%
≥ 8 mg/day	7	0.7% < 2.6 < 4.5%
TOTAL	270	

**Table 5 nutrients-11-02308-t005:** Repartition of consumers that have isoflavone intake below and over 45mg/day.

Isoflavone Consumption	<45 mg/day	≥45 mg/day	TOTAL
Apart from you who consume soy at yours?			
Nobody	43 (43%)	18 (47%)	61
My husband	25 (25%)	9 (24%)	34
My child below 3 years-old	1 (1%)	1 (2.6%)	2
My child over 3 years-old	4 (4%)	2 (5.3%)	6
My adolescent	9 (9%)	4 (10.5%)	13
Others	18 (18%)	4 (10.5%)	22
Total	100 (100%)	38 (100%)	138

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
