# Peer review of "New Evaluation of Isoflavone Exposure in the French Population"

_nutrients, 2019, doi:10.3390/nu11102308_

Round 1

Reviewer 1 Report

This manuscript is a well written article on interesting aspects of isofllavones.

Author Response

Thank you for your comment. We hope the attached file will answer to your expectancies.

Reviewer 2 Report

 The study relates the assessment of exposure to estrogenic isoflavones of French consumers. Authors have used two approaches. One through foodstuff-label analysis and isoflavone  measurements in diets, and another through a survey on premenopausal women. For the foodstuff approach, 150 items were analyzed for genistein and daidzein. The manuscript is easy to read and understand, and the results are clearly presented. Furthermore, the article is well constructed, the experiments were well conducted, and analysis was well performed.

Author Response

Thank you for your comments. We hope the attached file will answer to your expectancies.

Reviewer 3 Report

The authors undertook very interesting and important studies on the potential consumption of isoflavones by the French. Isoflavones are often described as being safe for health, with high soy intake in Asian countries given as evidence. However, in fact the amount of isoflavones consumed in Asia in the past may be significantly lower than their content in food products currently on the market. Knowing that isoflavones have significant biological activity and that soy is very widely used as a food additive, it is important to estimate the potential daily intake of isoflavones by humans. It may be necessary to implement procedures to remove isoflavones in food production processes.

I have a few questions and comments:

Line 21: Please round off the values to the hundredths after the decimal point. This note applies to the entire text of the publication, tables and supplement.

Line 31: The reference number [1] is larger than the rest of the text, please correct throughout the publication.

Line 32: “because these compounds play essential role in these plants” This statement is not an explanation for the high isoflavone content, please change it. Maybe “in which these compounds play essential role” or something similar.

Line 62: antiantogenic?

Figure 1, point 10: it should be” “Processed food in cans”

Line 121: I believe that the methodology for determining isoflavone content should be further described. In the rest of the text, it is worth highlighting what has been done. It was very laborious, and reading the publication I have the impression that this part of the research was left unsaid. Please delete reference 21, this is a review work and there is no ELISA method described.

Please review the literature carefully and correct minor errors.

Supplement:

Page 2: (all data are given in aglycone equivalen per weight) – please delete this

Page 4: Please review the literature carefully and correct minor errors. Latin names should be written in italic.

Page 15: Please add list of references included in the table.

Page 17: Please replace the dots with commas in the table.

The tables have both periods and commas at fractional values, please harmonize.

Page 21: What was the reason for choosing such meals (question 6)?

Despite the above remarks, I think that the work presented to me for review is very interesting and valuable.

Author Response

Thank you for your careful reading and your valuable comments that improved our essay. We hope the attached file will answer to your expectancies.
